# Continuous Renal Replacement Therapy with High Flow Rate Can Effectively, Safely, and Quickly Reduce Plasma Ammonia and Leucine Levels in Children

**DOI:** 10.3390/children6040053

**Published:** 2019-04-04

**Authors:** Fatih Aygun, Fatih Varol, Cigdem Aktuglu-Zeybek, Ertugrul Kiykim, Halit Cam

**Affiliations:** 1Department of Pediatric Intensive Care Unit, Istanbul University-Cerrahpasa, Cerrahpasa Medical Faculty, Fatih, Istanbul 34098, Turkey; dr_fvarol@yahoo.com (F.V.); hacam@istanbul.edu.tr (H.C.); 2Division of Nutrition and Metabolism, Department of Pediatrics, Cerrahpasa Medical Faculty, Istanbul University-Cerrahpasa, Istanbul 34098, Turkey; dracaz@istanbul.edu.tr (C.A.-Z.); ertugrulkiykim@hotmail.com (E.K.)

**Keywords:** mapple syrup urine disease, continuous renal replacement therapy, hyperammonemia, neonate

## Abstract

*Introduction*: Peritoneal dialysis and continuous renal replacement therapy (CRRT) are the most frequently used treatment modalities for acute kidney injury. CRRT is currently being used for the treatment of several non-renal indications, such as congenital metabolic diseases. CRRT can efficiently remove toxic metabolites and reverse the neurological symptoms quickly. However, there is not enough data for CRRT in children with metabolic diseases. Therefore, we aimed a retrospective study to describe the use of CRRT in metabolic diseases and its associated efficacy, complications, and outcomes. *Materials and Methods*: We performed a retrospective analysis of the records of all patients admitted in the pediatric intensive care unit (PICU) for CRRT treatment. *Results*: Between December 2014 and November 2018, 97 patients were eligible for the present study. The age distribution was between 2 days and 17 years, with a mean of 3.77 ± 4.71 years. There were 13 (36.1%) newborn with metabolic diseases. The patients were divided into two groups: CRRT for metabolic diseases and others. There was a significant relationship between the groups, including age (*p* ≤ 0.001), weight (*p* = 0.028), blood flow rate (*p* ≤ 0.001); dialysate rate (*p* ≤ 0.001), and replacement rate (*p* ≤ 0.001). The leucine reduction rate was 3.88 ± 3.65 (% per hour). The ammonia reduction rate was 4.94 ± 5.05 in the urea cycle disorder group and 5.02 ± 4.54 in the organic acidemia group. The overall survival rate was 88.9% in metabolic diseases with CRRT. *Conclusion*: In particularly hemodynamically unstable patients, CRRT can effectively and quickly reduce plasma ammonia and leucine.

## 1. Introduction

Peritoneal dialysis and continuous renal replacement therapy (CRRT) are the most frequently used treatment modalities for acute kidney injury in critically ill children. CRRT is currently being used for the treatment of several non-renal indications, such as metabolic acidosis and congenital metabolic diseases [1]. Because patients with inborn errors of metabolism have a high risk of hemodynamic instability due to low weight, usage of drugs such as arginine and lower systemic pressure etc., CRRT is accepted as a better choice than intermittent hemodialysis for treatment [2,3].

Continuous venovenous hemofiltration (CVVH), continuous venovenous hemodialysis (CVVHD), and hemodiafiltration (CVVHDF) have been used successfully in critically ill children since 1970s [4]. However, it is more widely used in the treatment of metabolic coma in recent years. CRRT is a treatment choice for metabolic coma in children, as it allows rapidly, continuous and programmed removal of toxic metabolites, which is tolerated better hemodynamically [5,6]. There are some differences from intermittent hemodialysis. With intermittent hemodialysis, clearance is limited by blood flow rates [7]. Furthermore, when intermittent HD is discontinued, there may be a rebound effect. However, much better clearance values and less rebound effect have been reported with CRRT [8]. In the literature, mortality, dialysis dependence, and effectivity were similar to these CRRT modalities (CVVH, CVVHD, and CVVHDF) [6,8].

During life-threatening acute metabolic crisis, plasma toxic metabolite concentrations like ammonia and branched amino acids should be reduced as soon as possible [2]. Hyperammonemia is observed in metabolic disorders such as urea cycle disorders (UCD), organic acidemias and fatty acid oxidation defects. Increased ammonia levels causing neurological symptoms can be detected in various inborn errors of metabolism (IEM). CRRT has been reported as an effective way of ammonia removal and reversal of neurological symptoms [9].

Maple syrup urine disease (MSUD) is an inherited disorder of branched chain amino acids (BCAA) caused by defective ketoacid dehydrogenase complex. Infants with classic MSUD, the most severe form, present within the first few days of life with irritability, poor feeding, lethargy, intermittent apnea, opisthotonus, abnormal movements, coma, and encephalopathy. The toxic accumulation of BCAA, mostly leucine, is associated with the appearance of neurological symptoms. Early treatment is essential to prevent neurotoxicity and death [9,10]. CRRT is a choice of treatment in metabolic crisis of MSUD resistant to conservative treatment modalities including nutritional support [1,11].

There is not enough data for CRRT in children with metabolic diseases. Also, the most appropriate and effective dialysis technique in the pediatric intensive care unit (PICU) has not been determined yet. The aim of this retrospective study is to describe the efficacy, complications and outcome of CRTT in IEMs.

## 2. Materials and Methods

### 2.1. Study Design

We conducted a single-center, retrospective study in a 7-bed PICU tertiary care hospital with two isolation rooms, 6 ventilators and two Prismaflex^TM^ hemofiltration machines (Baxter, Deerfield, IL, USA). The study included critically ill children 2 days-18 years of age who underwent CRRT from December 2014 to November 2018. Ethical approval for this study was obtained from the Local Ethics Committee (December 2018, No. 29430533-903.99.-92611).

### 2.2. Patient Population and Data Collection

We performed a retrospective analysis of the records of all patients admitted in the PICU between December 2014 and November 2018. Patients with a history of chronic renal failure and PICU stay duration of <24 h and those who died on the first day of admission were excluded. All data were obtained from electronic and written medical records in accordance with the ethical principles for medical research (Figure 1).

Demographic data and reason of hospitalization were recorded. The patients’ sex and age, underlying disease, invasive or non-invasive mechanical ventilation (NIV) requirement, inotropic drug usage, red blood cell transfusion, duration of hospitalization in the intensive care unit, mortality, plasma exchange, catheter diameter, and hemodialysis filters were recorded (Table 1).

### 2.3. Catheterization and Continuous Renal Replacement Therapy (CRRT)

The CRRT indications included fluid overload, electrolyte imbalance, metabolic acidosis, inherited metabolic diseases metabolic disease, and intoxication. Some patients had more than one indication. Double-lumen central venous catheters were percutaneously placed along the femoral vein, jugular vein, or subclavian vein by a single specialist. Primarily, jugular and femoral catheters were inserted due to the potential complications such as pneumothorax and hemothorax. The catheter was dilatated by the dilatator of the 4-Fr catheter and then by dilatators of the 7-Fr and 8-Fr catheters. We used 4 Fr catheters as guide and used the smallest possible hemodialysis catheter. The location of all jugular and subclavian catheters was controlled by chest X-ray. The right side of the patient was favored for the insertion of the jugular and subclavian catheters due to decreased complication risk.

All patients received sedation and analgesia prior to catheterization. Midazolam, ketamine, and remifentanil were preferred for sedation and analgesia. Lidocaine was used as a local anesthetic. Neuromuscular blockers were not used. Surgical catheters were not attached (cut down etc.). The aseptic method was used during catheterization. The entry location was sterilized with 1% povidone-iodine and left to dry. Disposable sterile covers that covered the whole body were used. A sterile apron, mask and bonnet were used. The catheters were attached with the Seldinger method and fixed to the skin with 2.0 sharp-pointed silk sutures. Ultrasonography (USG) was not used.

Anticoagulation was performed using unfractionated heparin. The dosage of heparin was adjusted on the basis of the activated partial thromboplastin time (aPTT), which was checked every 4 h and maintained at 60–80 s. Prismaflex^TM^ hemofiltration systems and poly membrane (AN69) filters were used. The blood flow rate was 5–20 mL/kg/min. The dialysate rate, replacement fluid rate, and ultrafiltration rate were adjusted on the basis of the patients’ diagnoses. The dialysate flow rates were set between 2 and 12.9 L/1.73 m^2^/h. Multibic^TM^ (Fresenius Medical Care, AG Co., Bad Hamburg, Germany) was used as dialysate solution. We used CVVHD and CVVHDF.

Patients with body weight less than 5 kg underwent blood priming and the system was subjected to internal hemodialysis for 10 min to prevent bradykinin release and electrolyte instability. The dialysis sets were filled with 5% albumin for patients having hemodynamic instability and body weighth >5 kg. The blood priming bag included red blood cells mixed 1:1 with normal saline.

We used unfractionated heparin for anticoagulation. The dosage of heparin was arranged according to partial thromboplastin time (aPTT) that was checked every 4 h, keeping aPTT in 60–80 s. Blood gas, ammonia, electrolyte, serum leucine, and coagulation controls were taken from all patients for every 4 h.

### 2.4. Support Treatment for Metabolic Diseases

All of the patients were given high energy to prevent the catabolic process. Intravenous thiamine supplementation was provided for patients with MSUD. For patients with hyperammonemi; arginine, citrulline, carboglu (carglumic acid) and sodium benzoate support were supplied. Lipid solution was started in all patients before the CRRT. All patients received intravenous phosphorus and calcium supplements.

### 2.5. Definition of Acute Kidney Injury

Acute kidney injury was defined as oliguria (urine output <0.5 mL per kg of body weight per h) and an elevated serum creatinine value for the patient’s age, or a 1.5-fold increase in the baseline creatinine level in 24 h. The estimated glomerular filtration rate was calculated according to the original Schwartz Formula. In addition, while serum urea and creatinine levels were measured, the glomerular filtration rate (GFR) was also controlled additionally which is a routine procedure of our laboratory.

### 2.6. CRRT for Metabolic Disease

In this study, an indication for CRRT was considered for ammonia concentrations >400 μmol/L and leucine concentrations >700 μmol/L. In addition, CRRT was started for metabolic diseases with unexplained encephalopathy.

The patients were divided into two groups: CRRT for metabolic disease and others. The groups were compared. The patients’ demographic, prognostic, and laboratory parameters in the admission were recorded. Laboratory parameters, and C-reactive protein, leukocyte, lymphocyte, platelet counts, and electrolytes in the first 24 h were considered as the possible risk factors of CRRT.

The patients with metabolic diseases were divided into three groups. These were MSUD, urea cycle disorder (UCD), and organic acidemias. The groups were compared.

### 2.7. Statistical Analysis

Statistical analysis was performed using the IBM SPSS 21.0 (SPSS Inc., Chicago, IL, USA). Continuous variables were expressed as means ± standard deviations and categorical variables as frequencies (percentages). Student’s *t*-test was used for continuous variables with a normal distribution. Pearson’s chi-square test and analysis of variance (ANOVA) were used for the comparison of the categorical data between the groups. A *p*-value of <0.05 was considered statistically significant.

## 3. Results

### 3.1. Demographics

Between December 2014 and November 2018, 97 patients were found eligible for the present study. Of the 97 patients, 55 (56.7%) were boys, and 42 (43.3%) were girls. The age distribution was between 2 days and 17 years, with a mean of 3.77 ± 4.71 years. The mean duration of hospitalization in the PICU was 8.61 ± 7.08 days.

Overall, metabolic decompensation/attack, septic shock, and respiratory failure were the main indications of CRRT. Inotropic drugs were used in 49 (50.5%) CRRT patients. MV was used in 41 (42.3%) CRRT patients, and NIV was efficient in 35 (36.1%) patients. Plasma exchange was performed in 23 (23.7%) patients. Fourteen (14.4%) CRRT patients died during their PICU stay. The demographic data of the CRRT patients are shown in Table 1.

Furthermore, 6.5-Fr (Able^®^, Oriontama Jaya, Indonesia) and 7-Fr (Medcomp^TM^, Harleysville, PA USA) hemodialysis catheters were used in 15 patients; 8-Fr hemodialysis catheters (Mahurkar^TM^, Medtronic, Minneapolis, MN, USA) in 62 patients; 10-Fr hemodialysis catheters (Mahurkar^TM^) in 9 patients; and 11.5-Fr hemodialysis catheters (Mahurkar^TM^) in 11 patients. Prismaflex^TM^ M60 (Baxter, Deerfield, IL, USA) used as the hemodialysis filter in 70 patients; Prismaflex^TM^, M100 in 14 patients; and Prismaflex^TM^ HF20 in 13 patients.

### 3.2. Comparison of the CRRT Usage with or Without Metabolic Disease in Pediatric Intensive Care Unit (PICU)

The patients were divided into two groups: CRRT for metabolic diseases and others. There was a significant relationship between these groups, including age (*p* ≤ 0.001), weight (*p* = 0.028), NIV (*p* = 0.028), red blood cell (RBC) transfusion (*p* = 0.005), plasma exchange (*p* ≤ 0.001), lymphocyte count (*p* = 0.008), platelet count (*p* ≤ 0.001), CRP (*p* = 0.008), and calcium level (*p* = 0.006). There was also a significant relationship between CRRT for metabolic disease and blood flow rate (*p* ≤ 0.001); dialysate rate (*p* ≤ 0.001); replacement rate (*p* ≤ 0.001); catheter sizes (*p* ≤ 0.001), and hemodialysis filters (*p* ≤ 0.001). There were 13 (36.1%) newborn who were used CRRT for metabolic diseases. There was no relationship in mortality between the groups. (Table 2).

The patients with metabolic diseases were divided into three groups: MSUD, UCD, and organic acidemia. There was a significant relationship between metabolic disease groups and demographic-prognostic factors, inotropic drug usage (*p* ≤ 0.001), NIV (*p* = 0.003), RBC transfusion (*p* = 0.039), duration of stay in the PICU (*p* ≤ 0.001), blood flow rate (*p* = 0.002), and replacement rate (*p* = 0.018) (Table 3).

The values of toxic metabolites were recorded before and after the procedure in patients who underwent CRRT due to metabolic diseases. The mean leucine level was 1777.0 ± 831.6 µmol/L at the beginning and 222.1 ± 143.5 µmol/L at the end of CRRT in the MSUD group. The mean ammonia level was 1143.5 ± 1069.9 µmol/L at the beginning and 50.0 ± 7.07 µmol/L at the end of the patients with UCD.

The ammonia reduction rate was 3.93 ± 3.68 (per hour) in the MSUD group, 4.94 ± 5.05 (per hour) in the UCD group, and 5.02 ± 4.54 (per hour) in the organic acidemia group. The leucine reduction rate was 3.88 ± 3.65 (% per hour) (Table 4).

Changes in serum ammonia levels during CRRT in patients with UCD are shown in Figure 2. There was rapid decrease in serum ammonia in the first hours, which was followed by a slow decline, as seen in the Figure 2. Six patients in the UCD group had serum ammonia level >1000 µmol/L, the decrease rate of ammonia level was faster in that group compared to the other UCD patients. Only two patients with UCD had ammonia level >500 µmol/L at the 12th hour.

### 3.3. Survival Analysis for Patients with CRRT Usage

Eighty-three (85.6%) patients were still alive after discharge from PICU. In the Kaplan–Meier analysis, there was no significant relationship between CRRT patients with or without metabolic disease (*p* = 0.414) (Figure 3). The mean overall survival time was 25.29 ± 1.69 days in CRRT patients with metabolic disease.

### 3.4. Complications in the Patients with Metabolic Disease

We experienced some complications during the CRRT performance in PICU. The most common complication was anemia and thrombocytopenia for newborns. Eight newborns developed thrombocytopenia that required transfusion during CRRT. Blood priming was applied to all patients having body weight less 5 kg and red blood cell transfusion was performed. None of the patients experienced serious life-threatening bleeding disorder such as intracranial or gastrointestinal.

Two patients with UCD died because of sepsis, they had sepsis on admission. One patient died due to thromboembolism that developed at the end of CRRT. There was no complication during catheter insertion. Catheters of patients with hyperammonemia were not removed after hemodialysis due to probable risk of rebound recurrence. No catheter-related infections or sepsis were observed.

In total 19 patients used to have inotrope during CRRT. Before starting CRRT, blood or albumin priming was performed in order not to impair blood pressure and hemodynamics. None of these complications were associated with high dialysate flow rate.

All the patients were followed by the pediatric nutrition and metabolism department, after discharged. In addition, concurrent follow-up was performed in the pediatric neurology department. The neurological outcome of patients with MSUD was good. However, patients with OCD and organic academia developed slight neurological influences. There were no patients with severe mental retardation and cerebral palsy.

## 4. Discussion

In our study, it was found that CRRT was effective and safe treatment of metabolic diseases in children. It was used in 36 (37.1%) patients for metabolic decompensates. Only four patients with metabolic diseases died. Although higher blood flow rate, dialysate rate and replacement rate were used in the metabolic group, serious complications were not detected during CRRT. MSUD patients required less inotropic drugs, had shorter PICU stay duration and none of the patients died during the follow up. There were no patients with severe mental retardation and cerebral palsy.

Several different metabolic diseases including urea cycle defects and organic acidemias can cause hyperammonemia in children [12]. Ammonia is an extreme neurotoxic metabolite in the human body. Prolonged persistence of elevated ammonia levels can cause astrocyte swelling, brain oedema, coma, severe disability, and even death. Thus, emergency treatment of hyperammonemia must be initiated before neurologic damage occurs [12,13]. The medical treatment of hyperammonemia begins with high-dose dextrose infusion to prevent catabolism. Cessation of protein consumption, higher caloric intake, insulin, carnitine, vitamins, and sodium benzoate/phenyl butyrate are used for treatment of hyperammonemia [14]. However, medical treatment modalities do not act sufficiently quickly to lower very high ammonia levels. Therefore, CRRT has been recommended to rapidly reduce the ammonia levels [7,11,15]. In the literature, the most important indicator of prognosis has been found to be the duration of hyperammonemia before the start of CRRT [16]. Currently, there is no consensus as to when it is appropriate to initiate dialysis in clinically significant hyperammonemia [17]. Indication for CRRT is considered for ammonia concentrations >400 μmol/L [2]. We used this indication in our study and 19 of 97 (19.6%) children received CRRT for hyperammonemia. We treated with sodium benzoate/phenyl butyrate, vitamins, high dextrose containing fluid (12–16 mg/kg/sc) with insulin and lipid infusion before the CRRT. CRRT was initiated, if the plasma ammonia level was >400 mmol/L despite treatment or if encephalopathy due to hyperammonemia developed.

In the literature, Picca et al. reported that CVVHD mode was the optimal modality for extracorporeal ammonium detoxification [15]. CVVHDF was used for all of our patients with hyperammonemia and our CRRT clearance was very high (5%). Interestingly, CRRT duration was longer in patients with organic acidemia. The reason for this was persistent metabolic (lactic) acidosis. Especially in two patients, CRRT was lasted more than 48 h. In our study, at first, we preferred CVVHD for MSUD, because MSUD is a benign disease compared with hyperammonemia. However, after 4 patients, we used CVVHDF because we thought it would be more effective. Therefore, we preferred CVVHD only in 4 patients. In the literature, there was no difference in survival when the patients were grouped according to the type of membrane or CRRT modality (CVVH, CVVHD or CVVHDF) [6,18]. In addition, mortality and dialysis dependence were similar between CVVH and CVVHDF groups [18]. Therefore, we used CVVHDF in our patients to benefit from both dialysate and filtration effect. The most important factor in our choice of CRRT was to prevent the rebound effect that is reported in the intermittent hemodialysis (HD). By contrast, McBryde et al. suggested that intermittent HD should be the first–line RRT modality for the treatment of metabolic diseases, with CRRT used to prevent rebound only after HD is discontinued [19].

The most important feature of our study that it was been used the higher rate of dialysate and replacement fluids during CRRT than others studies. We have used higher dialysate flow rate rather than the usual uses of acute kidney injury. Because, in a prospective study has demonstrated that slower reductions in ammonia levels may worsen long-term outcomes [20]. Therefore, we have kept dialysate rates higher than standard CRRT usage. Dialysate flow rate was 6083 ± 2690 mL/1.73 m^2^/h for the patients with UCDs and 5550 ± 2695 mL/1.73 m^2^/h for the patients with organic acidemias. High-flow dialysate rate was found very effective for reducing ammonia in our study. The ammonia reduction rates were almost 5% per h. This ratio seems a little low. We think there are two reasons for this. First, initial ammonia levels were <500 μmol/L in some patients. Second, in the first hours of dialysis, ammonia dropped much faster as seen in Figure 2.

In this study, we also slowly decreased the dialysis rate to avoid rebound upon the cessation of CRRT. There were a few rebound increases in plasma ammonia level after cessation of CRRT. None of our patients required CRRT due to rebound increase in ammonia level. Like our study, Hanudel et al. reported CRRT use consisting first of a dialysate flow rate of 5000 mL/h to rapidly decrease plasma ammonia levels in newborns [16]. In an another newborn study, it was revealed that high-dose CRRT for neonatal hyperammonemia was an effective and safe method for reducing ammonia [7].

We set higher dialysis speed for patients admitted with metabolic coma than other disease. To prevent electrolyte disorders that may arise from high flow dialysate rate, we monitored the serum biochemistry every 4 h. Multibic 4.0 dialysate solution containing 4 mmol of potassium was used in the patients with metabolic diseases. We did not encounter serious biochemical problems while monitoring our patients.

Increased plasma concentration of branched chain amino acids, mainly leucine, is associated with neurological symptoms and neonatal encephalopathy [10]. Peritoneal and intermittent hemodialysis are less effective and more dangerous than CRRT especially in low weight infants [21]. If the catabolic state of the patient continues at the end of dialysis, the leucine levels continue to increase due to the rebound effect of intermittent hemodialysis. CRRT may not lead to this rebound phenomenon because of longer duration use. In this study, we used high flow rate CRRT that rapidly reduced leucine levels. Our leucine removal rate was faster and safe like the literature [22]. By contrast with the hyperammonemia, there was no second session of CRRT used in our MSUD patients. In addition, duration of stay in the PICU was shorter than hyperammonemia. There was no died in our MSUD patients due to CRRT use. In addition, these patients had very good neurological outcomes.

Although CRRT is a life-saving treatment, it has some disadvantages, such as requirement of technical expertise, follow-up of hemodynamic and coagulation parameters, and need for vascular access and anticoagulation. There are some limitations in newborn CRRT such as vascular access, bleeding complications, and a lack of neonatal-specific hemofiltration devices. In our country, small-size catheters and filters are not accessible. Therefore, we used 8F catheters and large volume filters (0.6 m^2^). Blood priming was done before the starting the CRRT in our 13 newborn patients. A point to consider is that if blood priming is required, it may cause bradykinin release syndrome. Therefore, the CRRT machine was worked for 10 min for self-dialysis before being connected to the patient. In newborn patients, there were no complications when catheters were attached.

The most important complication in patients, especially newborns was low platelet count and anemia. Therefore, we prepared platelets and red blood cell suspensions for each patient before starting the dialysis. Coagulation with heparin was preferred. Citrate was not used in metabolic diseases. Our target for transfusion in the metabolic diseases was <10 g/dL for hemoglobin and <50 × 10^3^/µL for platelet count. Therefore, 31 of 36 patients were transfused. There was no severe bleeding in our patients because we closely followed vital signs, hematocrits and platelets. However, a patient with the diagnosis of UCD died as a result of thromboembolism, during the blood in the set being given back to the patient. In this study, blood pressure and other vital signs were followed closely. Therefore, blood pressures of our patients remained stable while CRRT use. In our study, most of the patients with UCD needed inotropic drugs (91.7%). This may be due to high ammonia values. In addition, before the CRRT, arginine and intravenous sodium benzoate/phenyl butyrate were used, which may have reduced blood pressure. In our study, the mean arterial blood pressure values were not compared. Because of the different age group of the patients, blood pressure percentile changes according to age.

In this study, the overall survival rate was 88.9% in metabolic diseases with CRRT. Also, neonatal patients’ survival rate was 84.6% (2/13 newborn). These rates were quite good, as in the literature [10]. Recently, Mok et al. reported CRRT mortality rate was 58% in newborns with CRRT [5]. Importantly, the success rate of the MSUD patients was 100% even though 6 newborns with CRRT use. In the CRRT patients without metabolic disease, the most important reason for high mortality could be sepsis and non-renal organ failures. In this group, eight patients (8/10) died during CRRT from refractory septic shock. In addition, 4 of these 8 patients were given the support of extracorporeal membrane oxygenation (ECMO).

## 5. Limitations of the Study

There are some limitations in the present study. It is a retrospective single-center and the long-term outcomes were not followed-up. The positive side of this study is that there are only a few published studies about CRRT for metabolic diseases in critically ill children. In addition, 36 patients were a relatively large population for metabolic diseases. Also, CRRT with low body weight and low age were other positive sides of this study.

## 6. Conclusions

Our results suggest that CRRT can effectively and quickly reduce plasma ammonia and leucine, particularly in hemodynamically unstable patients. Also, high dialysate flow rate does not increase the complications of CRRT in children. With an experienced team and close follow-up, CRRT is reliable and effective in critically ill patients, including those with low body weight.

## Figures and Tables

**Figure 1 children-06-00053-f001:**
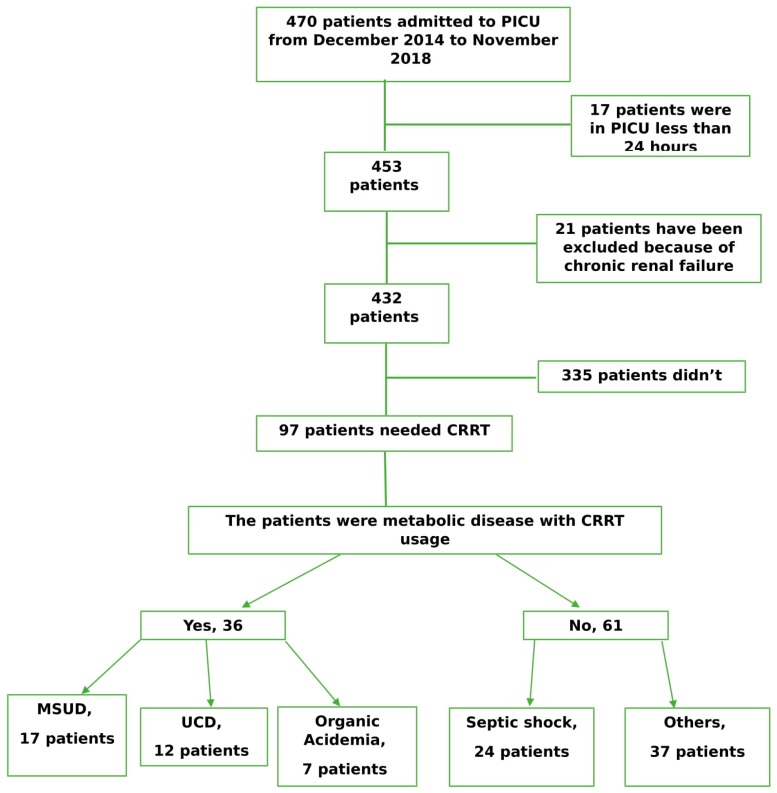
Cohort flow.

**Figure 2 children-06-00053-f002:**
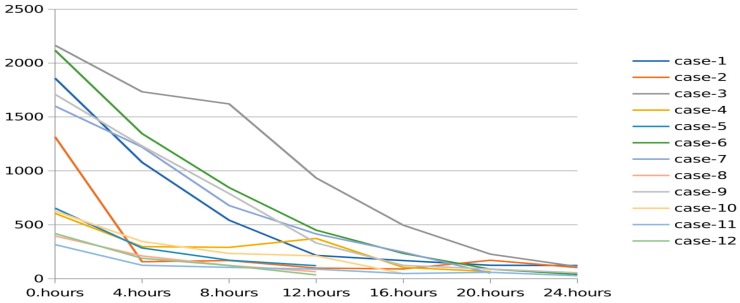
Changes in serum ammonia levels during CRRT in patients with urea cycle disorders.

**Figure 3 children-06-00053-f003:**
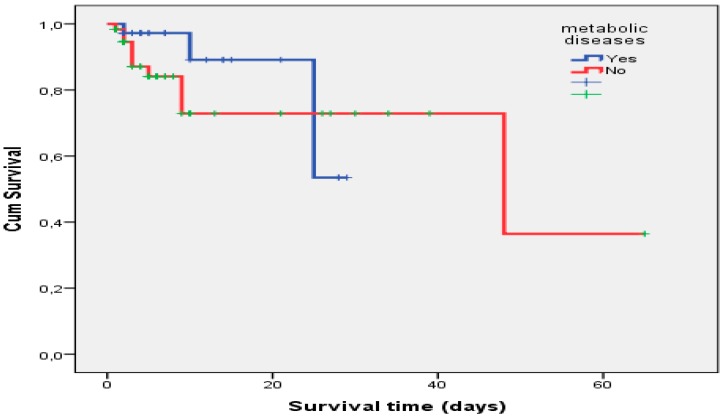
Kaplan–Meier survival curve of the patients with and without CRRT usage for metabolic diseases.

**Table 1 children-06-00053-t001:** Demographic characteristics of the CRRT ^1^ patients in the PICU ^2^.

Number of Patients	(*n* = 97)
**Gender**	*n* (%)
Male	55 (56.7%)
Female	42 (43.3%)
**Underlying Disease before the CRRT Use**	
Metabolic disease	36 (37.1%)
Septic shock	24 (24.7%)
Respiratory disease	10 (10.3%)
Neurological disease	4 (4.1%)
Cardiovascular Disease	4 (4.1%)
Intoxication	2 (2.1%)
Trauma	2 (2.1%)
Hematology-oncology-related diagnosis	3 (3.1%)
Acute renal pathology	12 (12.4%)
**Age (years)**	**Mean ± S.D.**
2 days–17.5 years	(4.15±4.91)
**Prognostic Factors**	*n* (%)
Red Blood Cell transfusion	67 (69.1%)
Inotropic medication	49 (50.5%)
Mechanical Ventilation	41 (42.3%)
NIV ^3^	35 (36.1%)
Plasma Exchange	23 (23.7%)
Death	14 (14.4%)
Duration of Stay in PICU (d)	8.61 ± 7.08
Duration of CRRT application (hours)	55.07 ± 79.62
**Catheters’ Diameter**	
6.5 and 7 Fr	15 (15.5%)
8 Fr	62 (63.9%)
10 Fr	9 (9.3%)
11.5	11 (11.3%)
**Hemodialysis Filter**	
0.2 m^2^	13 (13.4%)
0.6 m^2^	70 (72.2%)
0.9 m^2^	14 (14.4%)

^1^ Continuous renal replacement therapy; ^2^ patients in pediatric intensive care unit (PICU); ^3^ noninvasive mechanical ventilations.

**Table 2 children-06-00053-t002:** Comparison of clinical and laboratory findings for CRRT patients with and without metabolic disease.

Metabolic Disease with CRRT ^1^	Yes (*n* = 36) Mean ± S.D.	No (*n* = 61) Mean ± S.D.	*p*–Value
**Gender**			
Male	22 (61.1%)	33 (54.1%)	0.501
Female	14 (38.9%)	28 (45.9%)	
**Prognostic factors**			
Age (years)	1.01 ± 1.16	6.00 ± 5.32	≤0.001
Weight (kg)	9.62 ± 8.19	23.13 ± 21.30	0.028
Mechanical ventilation	17 (47.2%)	24 (39.3%)	0.448
Inotropic drug usage	19 (52.8%)	30 (49.2%)	0.732
Noninvasive mechanical ventilation	14 (38.9%)	21 (34.4%)	0.028
Red blood cell transfusion	31 (86.1%)	36 (59.0%)	0.005
Duration of stay in the PICU ^2^ (days)	8.86 ± 8.68	8.47 ± 12.37	0.867
Plasma exchange	3 (8.3%)	20 (32.8%)	≤0.001
Newborn	13 (36.1%)	0 (0%)	≤0.001
Mortality	4 (11.1%)	10 (16.4%)	0.474
Heart rate on admission (/min)	136.11 ± 22.56	147.00 ± 25.42	0.202
Respiratory rate on admission (/min)	35.33 ± 7.91	43.73 ± 12.57	0.012
**Laboratory findings**			
Leukocyte count (10^3^/µL)	9987.6 ± 5591.2	11584.4 ± 10569.6	0.514
Lymphocyte count (10^3^/µL)	4842.2 ± 3651.5	2397.2 ± 2824.4	0.008
Platelet count (10^3^/µL)	280000 ± 139449	157181 ± 113465	≤0.001
C-reactive protein level (mg/dL)	1.87 ± 2.99	5.79 ± 9.45	0.008
Sodium level (mmol/L)	137.30 ± 6.12	138.08 ± 9.08	0.721
Chlorine level (mmol/L)	98.96 ± 9.59	100.64 ± 11.21	0.591
Magnesium level (mg/dL)	1.87 ± 0.36	2.02 ± 0.34	0.301
Calcium level (mg/dL)	9.06 ± 0.92	8.18 ± 1.24	0.006
**Parameters and settings of CRRT**			
Duration of CRRT (hours)	36.42 ± 81.17	58.77 ± 52.14	0.104
Blood flow rate (mL/min)	6.67 ± 2.24	4.79 ± 0.58	≤0.001
Dialysate rate (mL/1.73 m^2^/h)	6076.03 ± 2383.95	1995.83 ± 32.28	≤0.001
Replacement rate (mL/kg/h)	45.06 ± 13.13	35.19 ± 6.32	≤0.001
**Catheter sizes**			
6.5 and 7 Fr	10 (27.8%)	5 (12.2%)	≤0.001
8 Fr	26 (72.2%)	36 (59.0%)	
10 Fr	0	9 (14.8%)	
11.5 and 12 Fr	0	11 (18.0%)	
**Hemodialysis filters**			
0.2 m^2^	11 (30.6%)	2 (3.3%)	≤0.001
0.6 m^2^	25 (69.4%)	45 (73.8%)	
0.9 m^2^	0	14 (22.9%)	
**Mod of CRRT**			
CVVHDF ^3^	32 (88.9%)	59 (96.7%)	0.122
CVVHD ^4^	4 (11.1%)	2 (3.3%)	

^1^ Continuous renal replacement therapy; ^2^ patients in pediatric intensive care unit (PICU); ^3^ continuous venovenous hemodiafiltration ^4^ continuous venovenous hemodialysis.

**Table 3 children-06-00053-t003:** Comparison of clinical findings in CRRT patients with metabolic disease.

		MSUD ^1^ (*n* = 17)	Urea Cycle Disorder (*n* = 12)	Organic Acidemia * (*n* = 7)	*p*-Value
Sex (male)		11 (64.7%)	8 (66.7%)	3 (42.9%)	0.541
Age (years)		1.25 ± 1.27	0.87 ± 1.18	0.68 ± 0.78	0.480
Weight (kg)		9.80 ± 11.04	9.35 ± 5.87	10.10 ± 8.77	0.988
Mechanical ventilation		5 (29.4%)	7 (58.3%)	5 (71.4%)	0.111
Inotropic drug usage		3 (17.6%)	11 (91.7%)	5 (71.4%)	≤0.001
Noninvasive mechanical ventilation		4 (23.5%)	3 (25.0%)	7 (100%)	0.003
Blood component transfusion		12 (70.6%)	12 (100%)	7 (100%)	0.039
Duration of stay in the PICU ^2^ (days)		3.94 ± 3.07	10.92 ± 7.76	17.29 ± 11.98	≤0.001
Mortality		0 (0%)	2 (16.7%)	2 (28.6%)	0.097
Duration of CRRT ^3^ (hours)		22.53 ± 17.84	19.42 ± 16.43	35.00 ± 13.30	0.144
Blood flow rate (ml/min)		7.81 ± 2.17	4.91 ± 0.94	7.25 ± 2.22	0.002
Dialysate rate (ml/1.73 m^2^/h)		6225 ± 2196	6083 ± 2690	5550 ± 2695	0.864
Replacement rate (ml/kg/h)		41.07 ± 13.42	53.17 ± 9.88	38.17 ± 11.23	0.018
**Age distribution**	0–28 days	6 (35.3%)	6 (50.0%)	1 (14.3%)	0.187
	28 days–2 years	7 (41.2%)	5 (41.7%)	6 (85.7%)	
	>2 years	4 (23.5%)	1 (8.3%)	0 (0%)	

^1^ MSUD: maple syrup urine disease, ^2^ PICU: pediatric intensive care unit, ^3^ CRRT: continuous renal replacement therapy, *: without MSUD.

**Table 4 children-06-00053-t004:** The changes of metabolites level in metabolic diseases when CRRT.

	MSUD ^1^ (*n* = 17)	UCD ^2^ (*n* = 12)	Organic Acidemia * (*n* = 7)
Ammonia (start the CRRT ^3^) (µmol/L)	164.8 ± 145.5	1143.5 ± 1069.9	538.6 ± 1017.9
Ammonia (end of the CRRT) (µmol/L)	57.7 ± 51.4	50.0 ± 7.07	71.23 ± 55.1
Ammonia reduction rate (% per hour)	3.93 ± 3.68	4.94 ± 5.05	5.02 ± 4.54
Leucine (start the CRRT) (µmol/L)	1777.0 ± 831.6	-	-
Leucine (end of the CRRT) (µmol/L)	222.1 ± 143.5	-	-
Leucine reduction rate (% per hour)	3.88 ± 3.65	-	-
Duration of CRRT (hours)	22.53 ± 17.84	19.42 ± 16.43	35.00 ± 13.30

^1^ Maple syrup urine disease, ^2^ urea cycle disorder, ^3^ continuous renal replacement therapy, * without MSUD.

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
