# Peer review of "Continuous Renal Replacement Therapy with High Flow Rate Can Effectively, Safely, and Quickly Reduce Plasma Ammonia and Leucine Levels in Children"

_children, 2019, doi:10.3390/children6040053_

Reviewer 1 Report

Dear Authors,

in your manuscript you describe the effectiveness of CRRT. Overall the manuscript is well written, however  there are few minor comments:

Abbreviations:

BCAA for branched chain amino acids should be used rather than BCHA

MV, NIV, CVVHD and CVVHDF should be defined on first use

subchapter 2.2; line 69: "The study was approved by local ethics committee."  - this sentence is a duplication and can be deleted in this subchapter

subchapter 2.4  - please provide a lit. citation for the Schwartz Formula

page 3; line 104 - statistical Analysis should be a separat Sub chapter 2.6

page 4; lines 125-129 - this paragraph should be moved to the materials & methods section.

subchapter 3.2 has a false heading, since it deals not exclusively with metabolic diseases

page 6; line 153 - it must be ammonia,(not ammoniac)

Author Response

March 23, 2019

Dear Reviewer;

I, along with my coauthors, would like to re-submit the attached manuscript entitled “Continuous Renal Replacement Therapy with High Flow Rate Can Effectively, Safely, and Quickly Reduce Plasma Ammonia and Leucine Levels in Children” as an Original Research article. The manuscript ID is MS No.: Children-454008

The manuscript has been carefully rechecked and appropriate changes have been made in accordance with the reviewers’ comments. The new changes are with yellow highlights. The responses to the reviewers’ comments have also been prepared. We hope that the revised manuscript is now suitable for publication in your journal.

I look forward to your reply.

Sincerely,

Fatih Aygun

Department of Pediatric Intensive Care Unit

Istanbul University Cerrahpasa Medical Faculty, 34098, Fatih, Istanbul, Turkey

Tel.: +90 (532)532 786 86 82; (530)5534513

Fax: +90 (212) 6328633

Reviewer 2 Report

The authors describe a retrospective analysis of continuous renal replacement therapy (CRRT) in one single center. They collect 97 cases and among them 36 have metabolic disease. The age and disease distribution of cases are quite wide. There are number of comments on the manuscript:

Introduction

The authors say that “-- -- CRRT is accepted as a better choice than intermittent hemodialysis for treatment [2,3].”. These are very old references, and the authors should consult more recent published guidelines

The authors say that “There is not enough data for CRRT in children with metabolic diseases. Also, the most appropriate and effective dialysis technique in PICU has not been determined yet.”. Again, the authors should make these conclusions after consulting more recently published guidelines.

M&M

From the description in the methods, the center is not equipped with hemodialysis. There should be cases who need more aggressive metabolic correction than using CRRT. The authors should comment on the fact that hemodialysis is not available or not considered in the center

The sentence “No surgical catheters were attached.” is not clear

The authors do not describe the procedures of CRRT, for example, CVVH or CAVH?

Result

“Fourteen (14.4%) CRRT patients died during their PICU stay” is a quite high mortality rate. Is it related to CRRT?

36 cases with a variety of metabolic disease is actually not a great number.

“The mean leucine level was 1777.0 ± 831.6 μmol/L at the beginning and 222.1 ± 143.5 μmol/L at the end of CRRT in the MSUD group. The mean ammoniac level was 1143.5 ± 1069.9 μmol/L at the beginning and 50.0 ± 7.07 μmol/L at the end of the patients with UCD.” This looks beautiful. But an ammoniac reduction rate 4.94 ± 5.05 (percent ?  per hour) in the UCD group may not be good enough. This will mean that a patient with an initial ammonia level of 2000 umol/L will still have a level of >1000 10 hours after CRRT, or still >500 one day after CRRT.

Author Response

(The authors gave the same response as above.)

Reviewer 3 Report

This study addresses an important topic and reports new comprehension of the use of CRRT  in critically ill children with metabolic disease. It deserves to be published. However, there are some comments:

  Material and methods: Patient population and data collection section could be described as a flowchart of exclusion criteria and outcomes.

Catheterization and CRRT section: Would authors delineate what criteria they use for choosing the CV catheters.  Were there any side effects, especially with the subclavian vein catheterization? Are there data what SCV vein, left or right, have been used? Have any modalities been used to confirm the location of the SCV catheter?

The authors calculated the estimated GFR (eGFR) using the original Schwartz formula, which does not incorporate the body weight, directly affecting GFR. Body weights could provide more accurate assessment of the kidney damage. It would be good if the authors provide these data.

Did the authors estimate the creatinine clearance, which takes into consideration the body weight?

 Results: There are no figures (plots, histograms and etc.) in this paper. It is difficult to read data from tables only. Graphically presented results would improve the perception from reading these data.

CRRT can develop different cardiovascular abnormalities. Authors mention that they monitored blood pressure. Would authors provide blood pressure data, ECG data etc., in the supplemental section if this exceeds word/figure limit?

Discussion: Lines: 163 – 167: Authors discussed follow-ups, however it is unclear when  patients were followed up on. Was it for a defined period of time after treatment? Would authors provide these data, preferably in a graph form? Also would authors provide the Kaplan-Meier survival analysis?

Line 193: “no serious side effects were detected”. What were the side effects you were looking to detect?

Author Response

March 23, 2019

Dear Reviewer;

I, along with my coauthors, would like to re-submit the attached manuscript entitled “Continuous Renal Replacement Therapy with High Flow Rate Can Effectively, Safely, and Quickly Reduce Plasma Ammonia and Leucine Levels in Children” as an Original Research article. The manuscript ID is MS No.: Children-454008

The manuscript has been carefully rechecked and appropriate changes have been made in accordance with the reviewers’ comments. The new changes are with yellow highlights. The responses to the reviewers’ comments have also been prepared. We hope that the revised manuscript is now suitable for publication in your journal.

I look forward to your reply.

Sincerely,

Fatih Aygun

Department of Pediatric Intensive Care Unit

Istanbul University Cerrahpasa Medical Faculty, 34098, Fatih, Istanbul, Turkey

Tel.: +90 (532)532 786 86 82; (530)5534513

Fax: +90 (212) 6328633

Round  2

Reviewer 2 Report

The manuscript improves significantly in the parts of descriptions about the method and the results. However, the background and discussion parts are still not adequate (this comment apply to the abstract, introduction, and discussion). The authors must present the newest guidelines about the selections of HD, CVVD, and CVVHD in the background/introduction. Then discuss about these guidelines by their own experiences in the discussion. 

Author Response

March 30, 2019

Dear Reviewer;

I, along with my coauthors, would like to re-submit the attached manuscript entitled “Continuous Renal Replacement Therapy with High Flow Rate Can Effectively, Safely, and Quickly Reduce Plasma Ammonia and Leucine Levels in Children” as an Original Research article. The manuscript ID is MS No.: Children-454008

The manuscript has been carefully rechecked and appropriate changes have been made in accordance with the reviewers’ comments. The newer changes are now with green highlights while previous revision is with yellow highlights. The responses to the reviewers’ comments have also been prepared. We hope that the revised manuscript is now suitable for publication in your journal.

I look forward to your reply.

Sincerely,

Reviewer 3 Report

none

Author Response

(The authors gave the same response as above.)
